# Epigenetic insights into GABAergic development in Dravet Syndrome iPSC and therapeutic implications

Jens Schuster[†], Xi Lu[†], Yonglong Dang[†], Joakim Klar, Amelie Wenz, Niklas Dahl*, Xingqi Chen*

Department of Immunology, Genetics and Pathology, Uppsala University and Science for Life Laboratory, Uppsala, Sweden

*For correspondence:
niklas.dahl@igp.uu.se (ND);
xingqichen2015@gmail.com (XC)

[†]These authors contributed equally to this work

Competing interest: The authors declare that no competing interests exist.

**Abstract** Dravet syndrome (DS) is a devastating early-onset refractory epilepsy syndrome caused by variants in the *SCN1A* gene. A disturbed GABAergic interneuron function is implicated in the progression to DS but the underlying developmental and pathophysiological mechanisms remain elusive, in particularly at the chromatin level. Induced pluripotent stem cells (iPSCs) derived from DS cases and healthy donors were used to model disease-associated epigenetic abnormalities of GABAergic development. Chromatin accessibility was assessed at multiple time points (Day 0, Day 19, Day 35, and Day 65) of GABAergic differentiation. Additionally, the effects of the commonly used anti-seizure drug valproic acid (VPA) on chromatin accessibility were elucidated in GABAergic cells. The distinct dynamics in the chromatin profile of DS iPSC predicted accelerated early GABAergic development, evident at D19, and diverged further from the pattern in control iPSC with continued differentiation, indicating a disrupted GABAergic maturation. Exposure to VPA at D65 reshaped the chromatin landscape at a variable extent in different iPSC-lines and rescued the observed dysfunctional development of some DS iPSC-GABA. The comprehensive investigation on the chromatin landscape of GABAergic differentiation in DS-patient iPSC offers valuable insights into the epigenetic dysregulations associated with interneuronal dysfunction in DS. Moreover, the detailed analysis of the chromatin changes induced by VPA in iPSC-GABA holds the potential to improve the development of personalized and targeted anti-epileptic therapies.

## eLife assessment

This is a potentially **useful** study that shows changes in the chromatin landscape of GABAergic neurons in induced pluripotent stem cells (iPSCs) derived from both Dravet Syndrome (DS) patients and healthy donors. The strength of the evidence is currently **incomplete** because the authors compared iPSCs from different individuals, rather than isogenic controls. A strategy for minimizing variability across cell lines is used, but the explanation is not complete. The revised manuscript adds RNAseq and qPCR measurements of the expression of the gene SCN1A, however these do not appear to agree, perhaps because of the way the qPCR measurements are normalized, and there is no measurement of Nav1.1, the gene product thought to be responsible for the majority of DS cases. Hence the evidence that there is reduced expression of SCN1A or its gene product is not complete and therefore it is difficult to evaluate whether or not the observed epigenetic changes are causal. The work would potentially be of interest to scientists who study development, developmental disorders, and epigenetic contributions to disease.

## Introduction

DS is an early onset and intractable epilepsy with an unfavorable long-term outcome. The first seizures are usually triggered by fever within the first 12 months of life (*He et al., 2022*; *Dravet, 2011*). Characteristic clinical features are age-related progression of seizures, cognitive decline, behavioral problems, and movement disorder. The complex neurological symptoms associated with DS suggest underlying pathophysiological mechanisms that interfere with brain development (*He et al., 2022*; *Dravet, 2011*; *Li et al., 2021*; *Ragona, 2011*). Approximately 80% of DS cases carry heterozygous variants in the *SCN1A* gene encoding the α-subunit voltage-gated sodium channel (Nav1.1) (*He et al., 2022*; *Brunklaus and Zuberi, 2014*). Prior insights on the neuropathophysiology in DS have come from the *Scn1a* heterozygous mice, indicating a vital role of cortical interneurons in the pathogenesis. Mice with Nav1.1 haploinsufficiency in GABAergic interneurons exhibit spontaneous seizures and behavioral abnormalities (*Cheah et al., 2012*; *Kalume et al., 2015*; *Ito et al., 2013*; *Han et al., 2012*) that are associated with subnormal sodium currents and impaired excitability of inhibitory interneurons (*Yu et al., 2006*; *Tai et al., 2014*). Additionally, human iPSC-derived neural cells carrying distinct pathogenic *SCN1A* variants have recapitulated electrophysiological and molecular perturbations in DS (*Sun et al., 2016*; *Schuster et al., 2019b*; *Maeda et al., 2016*; *Jiao et al., 2013*; *Higurashi et al., 2013*). In combination, these studies have confirmed that *SCN1A* variants cause delayed sodium currents in activated and mainly inhibitory GABAergic interneurons, supporting the notion that seizures in DS are caused by deficient cortical inhibition (*Sun et al., 2016*).

Interneuronal progenitors emerge from the embryonic subpallium, mainly the median ganglionic eminence, and migrate tangentially to cortical progenitor zones. The second half of gestation is a period of rapid development of the human cortical GABAergic system that continues after term (*Arshad et al., 2016*; *Xu et al., 2011*). The extended developmental processes into mature GABAergic interneurons require continuous and complex remodeling of the chromatin structure for transcriptional adaptations (*Allaway et al., 2021*). While it is now clear that the distinct features in DS are associated with a reduced sodium current density in Nav1.1 haploinsufficient interneurons and a disinhibition of the cortical network (*Cheah et al., 2012*; *Schuster et al., 2019b*; *Ogiwara et al., 2013*), the underlying epigenetic mechanisms are poorly understood.

Treatment of DS patients is challenging and seizure control is rarely attainable (*Aras et al., 2015*). The most frequently used first-line drug is valproic acid (VPA) (*He et al., 2022*). However, more than 50% of DS patients are drug resistant without any decrease in seizure-frequency upon treatment (*He et al., 2022*), indicating endogenous and subject-specific responses to VPA. The mechanisms by which VPA acts are not fully understood, but studies suggest that the drug inhibits histone deacetylase (HDAC) with an impact on chromatin structure and gene expression (*Ghodke-Puranik et al., 2013*; *Ximenes et al., 2012*). Furthermore, a direct interaction of VPA with sodium- or calcium-gated ion channels (e.g. Nav1.1) and enzymes crucial for GABA turnover has been proposed (*Ghodke-Puranik et al., 2013*). While these studies have brought important information on the mode of action of VPA (*Balasubramanian et al., 2019*; *Jiang et al., 2019*; *Wang and Li, 2019*; *Golla et al., 2016*; *Balasubramanian et al., 2015*), the specific effects of VPA on the chromatin architecture of inhibitory GABAergic interneurons have not yet been investigated to the knowledge.

iPSC lines, carrying distinct disease-causing variants in the *SNC1A* gene and modeled GABAergic interneuron development in DS (*Schuster et al., 2019b*; *Schuster et al., 2019a*), were recently established. The DS-patient iPSC-derived GABAergic neurons recapitulated electrophysiological abnormalities and gene expression analysis showed enrichment for histone modifications, suggesting epigenetic abnormalities (*Schuster et al., 2019b*). The dynamics of open chromatin in the DS-model of GABAergic development in DS were investigated by using the assay of transposase accessible chromatin sequencing (ATAC-Seq). The methodology has previously been used to investigate chromatin accessibility of annotated genes during brain development (*Inoue et al., 2019*; *Trevino et al., 2020*; *de la Torre-Ubieta et al., 2018*) and, specifically, in interneuronal differentiation (*Allaway et al., 2021*; *Inglis et al., 2020*). Herein, ATAC-Seq was employed to investigate the dynamics of chromatin accessibility in iPSC of DS-patients and healthy donors at four different time points (Day 0, Day 19, Day 35, and Day 65) of GABAergic development. Moreover, the effect of VPA on chromatin accessibility in GABAergic neurons was explored.

Accelerated chromatin changes in DS patient cells during the initial phase of GABAergic development (up to Day 19) were revealed when compared to cells of healthy donors. Further differentiation

revealed that DS-patient GABAergic cells acquire a distinct chromatin profile when compared to that of control cells. Notably, VPA treatment of GABAergic neurons leads to unspecific genome-wide changes in chromatin architecture that predict a promoted GABAergic development in some iPSC lines. The comprehensive investigation of the chromatin dynamics of GABAergic development in DS-patient iPSC sheds light on the underlying epigenetic abnormalities. Moreover, the in-depth characterization of chromatin changes in GABAergic neurons induced by VPA may bring important information for the development of individualized anti-seizure therapies.

## Results

### Dynamic chromatin accessibility in an iPSC model of GABAergic development

A protocol for GABAergic interneuronal differentiation of iPSC (*Schuster et al., 2019b*; *Schuster et al., 2019a*) was previously established. Analysis of RNAseq data at D19 and D65 of GABAergic differentiation revealed *SCN1A* expression in neuronal cells derived from healthy donors and from patients with Dravet disease (*Figure 1—figure supplement 1A*). The *SCN1A* expression was confirmed with qPCR at D19, D35, and D65 of differentiation whereas no expression could be detected in undifferentiated iPSC (D0; *Figure 1—figure supplement 1B*). ATAC-Seq was applied to investigate the dynamics of chromatin accessibility in GABAergic development associated with *SCN1A* variants. Firstly, a reference was obtained for chromatin accessibility changes during GABAergic interneuronal differentiation in iPSC from three healthy donors (Ctl) using ATAC-Seq (*Maroof et al., 2013*; *Figure 1A*). The differentiating cultures were collected at four distinct time points of differentiation at day 0 (D0; termed iPSC), D19 (neural progenitor cells; NPC), D35 (intermediate neuronal cells; imN), and D65 (GABAergic interneurons; GABA). The cultures displayed expression of marker genes confirming the development into GABAergic interneurons (*Schuster et al., 2019b*). As previously shown, cultures stained positive for pluripotent stem cell markers such as NANOG, SOX2, and OCT4 at D0. The expression of these markers decreased upon neural induction and disappeared with further differentiation. Markers for neural progenitor cells (FOXG1, PAX6) were expressed at D19 followed by the expression of immature neuronal cell markers (DCX and NKX2.1) at D35 and of markers for GABAergic interneurons (GAD1, TUBB3) at D65 (*Figure 1—figure supplement 1C* and *Schuster et al., 2019bSchuster et al., 2019b*).

The quality of ATAC-Seq data from each sample was validated by sequencing depth comparison, genome-wide correlation between technical replicates, and by calculating the fraction of reads in enriched peaks (FRiP), the transcription start site enrichment score (*Figure 1—figure supplement 2A–D*). Accessible chromatin peaks were captured at each time-point of differentiation: 58,382 peaks at D0 (Ctl-iPSC), 68,002 peaks at D19 (Ctl-NPC), 52,933 peaks at D35 (Ctl-iMN), and 74,561 peaks at D65 (Ctl-GABA) (*Supplementary file 1*). Genomic annotation of the accessible chromatin sites with respect to promoters, introns, and exons revealed a heterogenous distribution in iPSC, NPC, imN, and GABA (*Figure 1—figure supplement 2E*). Furthermore, principal component analysis (PCA) of chromatin accessibility at the four-time points demonstrated that the three biological replicates separated into four clusters corresponding to each of the four-time points of GABAergic development (*Figure 1—figure supplement 2F*), albeit with some degree of variability at the intermediate time points (D19 (NPC) and D35 (imN)), possibly reflecting cell line specific and endogenous differences reported previously (*Strano et al., 2020*). Therefore, the three Ctl-iPSC lines were treated as biological replicates for downstream analysis. Genome-wide correlation of ATAC-Seq with the previously published RNA-Seq data was also performed at D19 and D65 (*Figure 1—figure supplement 2G*). The correlation ranged between 0.52 and 0.57, further indicating the good quality of the ATAC-Seq data.

Next, changes in chromatin accessibility during GABAergic development in Ctl-iPSC were characterized. The ATAC-Seq peaks in iPSC (D0) were used as reference and compared to the accessible peaks in NPC, imN, and GABA. To avoid capturing the dynamic changes of accessible regions caused by variability across individuals, the dynamic changes of chromatin accessibility cell line by cell line across differentiation were initially compared. Subsequently, the common changes were extracted observed and across different cell lines at each time point (Methods). In total, 19,931 significant differential peaks (|log2(fold change (FC))|>1, false discovery rate (FDR)<0.01) were identified (*Figure 1—figure supplement 3A*, Methods). Next, an unbiased cluster method was employed to investigate the dynamics of these differentially accessible peaks, resulting in the identification

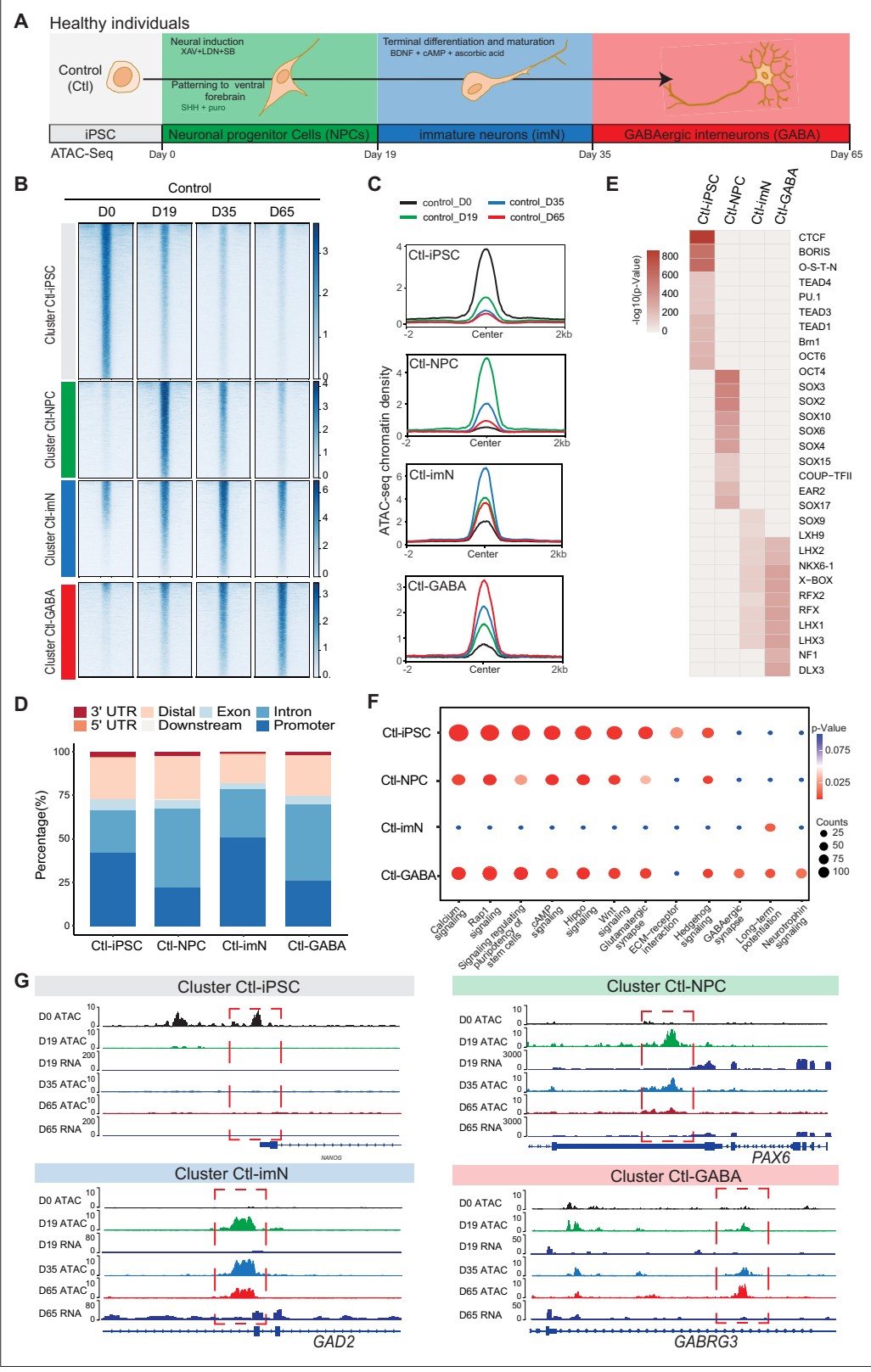

**Figure 1.** Chromatin accessibility dynamics during GABAeric interneuron differentiation of control induced pluripotent stem cells (iPSCs). (**A**) Schematic illustration of GABAergic interneuron differentiation protocol. Cells from iPSC differentiation were collected for assay of transposase accessible chromatin sequencing (ATAC-Seq). Interneuron differentiation was grouped into four stages: iPSC (Day 0), neural progenitor cells; NPCs (Day 19),

*Figure 1 continued on next page*

*Figure 1 continued*

immature neurons, imN (Day 35), GABAergic interneurons, GABA (Day 65). (**B**) Heatmap of four Clusters identified by time series analysis using temporal changes in chromatin accessibility compared to D0 (Day 0) and other points. The signal strengths were denoted by color intensities. (**C**) Line plots showing chromatin accessibility at cluster-specific regions from each time point. (**D**) Barplot for genomic distribution of differential chromatin accessible regions for each cluster. (**E**) Heatmap for top 10 transcription factors (TFs) enrichment in each cluster. The significance was denoted by color intensities. O-S-E-N=OCT4-SOX2-TCF-NANOG. (**F**) Bubble plot of Kyoto Encyclopedia of Genes and Genomes (KEGG) pathway enrichment for each cluster. p-value and enrichment were indicated. The corresponding comprehensive list of enrichment terms can be found in **Supplementary file 2**. (**G**) Genome browser view showing representative differential chromatin accessible regions at the indicated gene loci. Additionally, RNA-seq data (**Schuster et al., 2019b**) is visualized at D19 and D65.

The online version of this article includes the following figure supplement(s) for figure 1:

**Figure supplement 1.** validation of GABAergic differentiation.

**Figure supplement 2.** Quality control of assay of transposase accessible chromatin sequencing (ATAC-Seq) samples during GABAergic interneuron differentiation of Ctl-iPSC.

**Figure supplement 3.** Chromatin accessibility cluster identification during GABAergic interneuron differentiation of Ctl-iPSC.

of four distinct clusters (Cluster 1–4) (**Figure 1B**, **Figure 1—figure supplement 3B–D**). Cluster 1 (n=9565) consisted of peaks that were open D0 (cluster Ctl-iPSC) and that gradually closed during differentiation. Cluster 2 (n=3285) contained peaks that were predominantly open at D19 (cluster Ctl-NPC). Cluster 3 (n=3533) exhibited a high chromatin accessibility at both D19 and D35 (cluster Ctl-imN) but more closed at D65. Cluster 4 (n=3545) displayed the highest accessibility specifically at D65 (cluster Ctl-GABA; **Figure 1B and C**). A genomic annotation of the differential peaks in the four clusters with respect to promoters, exons, introns, and distal regulatory sequences revealed differences mainly in introns and promoter regions. The Ctl-NPC cluster and Ctl-GABA cluster showed the highest proportion of open chromatin in introns (45.6% and 43.4%, respectively) whereas in the Ctl-iPSC and Ctl-imN clusters, most accessible peaks were located in promoter regions (41.9% and 50.8%, respectively; **Figure 1D**).

Next, the transcription factor (TF) enrichment analysis was performed on the differential peaks specific to each cluster (**Figure 1E**). The Ctl-iPSC cluster showed specific enrichment for motifs of TFs for pluripotency, such as BRN1, and OCT6, among others (**Figure 1E**), whereas motifs for the SOX family of TFs were enriched in the Ctl-NPC-cluster. Ctl-NPC-specific transcription factor motifs (e.g. SOX family TFs) were detected in the NPC-specific cluster. Notably, motifs for common TFs (e.g. LHX family TFs, NKX6, RFX) as well as unique TFs (e.g. SOX9, NF1, and DLX3) were enriched in both the Ctl-imN-specific cluster and the Ctl-GABA specific cluster. The SOX family of TFs are important regulators of neuronal development (**Stevanovic et al., 2021**) and DLX factors function in orchestrating transcriptional activity during cortical GABAergic development (**Lindtner et al., 2019**). The temporal pattern of accessible and enriched TF motifs in the differentiating Ctl-iPSC cultures is thus consistent with the TFs required for neuronal and cortical GABAergic development in vivo, confirming the model system to be relevant.

Furthermore, the biological processes associated with each cluster was examined by assigning each differential peak to the nearest gene. The annotated genes were then used for KEGG analysis of each cluster (**Figure 1F**, **Supplementary file 2**). In the Ctl-iPSC cluster, enrichment of pathways related to pluripotency, such as signaling regulating pluripotency of stem cells was identified, while the Ctl-GABA cluster showed exclusive enrichment for pathways for GABAergic neurons, such as GABAergic synapse and Neurotrophin signaling (**Figure 1F**). Genome traces of annotated genes in each cluster validated the differential accessible chromatin, e.g., for *NANOG* in the Ctl-iPSC cluster, *PAX6*, and *GAD2* in the Ctl-NPC and Ctl-imN clusters, respectively, and *GABRG3* in the Ctl-GABA cluster (**Figure 1G**). Gene expression analysis of RNA-seq data at D19 and D65, as well as with qPCR at D0, D19, D35, and D65, indicated that expression levels of the four genes were consistent with the chromatin accessibility patterns (**Figure 1—figure supplement 3E**). Taken together, these findings show that the captured chromatin dynamics comply with the induction and development of GABAergic neurons.

## Dynamic chromatin accessibility in a DS-iPSC model of GABAergic development

Next, to gain a deeper understanding of chromatin dynamics during GABAergic development associated with DS, the same strategy was used as above for differentiation and ATAC-Seq analysis of iPSC-lines derived from the three DS patients (*Figure 2A*, *Figure 2—figure supplement 1*). Unfortunately, one of the DS-iPSC lines (DD4A) did not differentiate up to D65 for unknown reasons why data at the specific time-point are based on two DS-iPSC lines.

The ATAC-Seq data showed high quality and reproducibility at all four time-points (*Figure 2—figure supplement 1A–D*). In total, 89,363 peaks in DS-iPSC, 79,270 peaks in DS-NPC, 113,519 peaks in DS-imN, and 41,665 peaks in DS-GABA were detected (*Supplementary file 3*). The genomic distribution of these accessible chromatin peaks with respect to promoters, introns, and exons showed deviations when compared to Ctl-iPSC lines, mainly at D35 (DS-imN) and at D65 (DS-GABA) (*Figure 2—figure supplement 1E*). Furthermore, the PCA of chromatin accessibility in DS-iPSC replicates showed that they clustered together at D0, D19, and D35 (*Figure 2—figure supplement 1F*). Notably, the clusters at D19 (iPSC-NPC) and at D35 (iPSC-imN) showed a strong overlap in contrast to the well-distributed clusters at all four time points in Ctl-iPSC lines (*Figure 1—figure supplement 2F* and *Figure 2—figure supplement 1F*). Genome-wide correlation of ATAC-Seq and RNA-seq data at D19 and D65 showed a good correlation (0.49–0.55; *Figure 2—figure supplement 1G*). The different dynamics of chromatin accessibility of DS-iPSC lines when compared to that of Ctl-iPSC lines thus suggest a disrupted developmental trajectory into DS-GABAergic interneurons consistent with an altered function of inhibitory GABAergic interneurons in *Scn1a* heterozygous mice.

To further characterize chromatin changes in GABAergic differentiation of DS-iPSC lines, the same strategy was used as for Ctl-iPSC lines by using the profile of chromatin accessibility in DS-iPSCs (D0) as a reference. To avoid capturing the dynamic changes of accessible regions caused by variability across individuals, the same strategy was applied as described for the control samples (Methods). In total, 19,896 differential peaks were identified ($|log2(FC)|>1$, FDR <0.01) (Methods) that were clustered into four groups using an unbiased approach. The four DS-specific clusters showed similar patterns as those of the Ctl group. The number of accessible peaks were 8517 at D0 (DS-iPSC), 4317 at D19 (DS-NPC), 3579 at D35 (DS-imN), and 3483 at D65 (DS-GABA) (*Figure 2B and C*, *Figure 2—figure supplement 2A–D*). The differences in the genomic distribution of the DS-iPSC clusters were identified when compared to that in the Ctl-iPSC clusters (*Figures 1D and 2D*). In the control group, 42.0% of the peaks were located in promoter regions, whereas in the patient group, the proportion was slightly lower at 37.3%. Additionally, a slightly higher proportion of DS-iPSC-specific peaks were found in intron regions (26.5%) compared to Ctl-iPSC (24.6%). The genomic features of the DS-NPC-specific cluster were similar between the Ctl group and the DS group. In the DS-imN-specific cluster, a lower proportion of peaks were located in promoter regions when compared to that of the Ctl group (50.8% vs 23.1%), while a lower proportion of peaks were from intron regions in the Ctl group (27.8%) when compared to the DS group (46.2%). In the DS-GABA and Ctl-GABA specific clusters, 34.7% and 25.9% of the peaks were located in promoter regions, respectively, while 38.7% and 43.4% of the peaks were in intron regions in the Ctl and DS groups, respectively.

Next, the enrichment of motifs for trans-acting TFs within the differential peaks of each DS-cluster at each time-point was examined (*Figure 2E*). The DS-iPSC cluster-specific peaks showed enrichment of motifs from pluripotency-related TFs, such as BRN1 and OCT4, whereas TF motifs for the SOX family of TFs were enriched in both the DS-iPSC cluster and DS-NPC cluster (*Figure 2E*). Peaks at TF motifs for LHX3 and NKX, specific for GABAergic neurons were enriched only in the DS-NPC cluster but not in the DS-GABA cluster. However, the DS-GABA-specific peaks were enriched at motifs for the activator protein-1 (AP-1) family of TFs, including JUN, FRA, and FOSL. Enriched motifs for the BACH2 TF were observed only in the DS-GABA specific peaks and not in the Ctl-GABA peaks. Together, these data further support that DS-iPSC, although they initially respond to the protocol for GABAergic induction with accessible motifs for the SOX family of TFs, have a disrupted trajectory into GABAergic interneurons as shown by the loss of motifs for DLX and LHX at D65.

Next, the differential accessible peaks were annotated to nearby genes and performed KEGG pathway analysis for each cluster (*Figure 2F*, *Supplementary file 4*). Similar to Ctl-iPSC, the peaks of the DS-iPSC-specific cluster were enriched for pathways related to pluripotency. However, and in contrast to the control group, pathways related to GABAergic synapse were enriched already in the

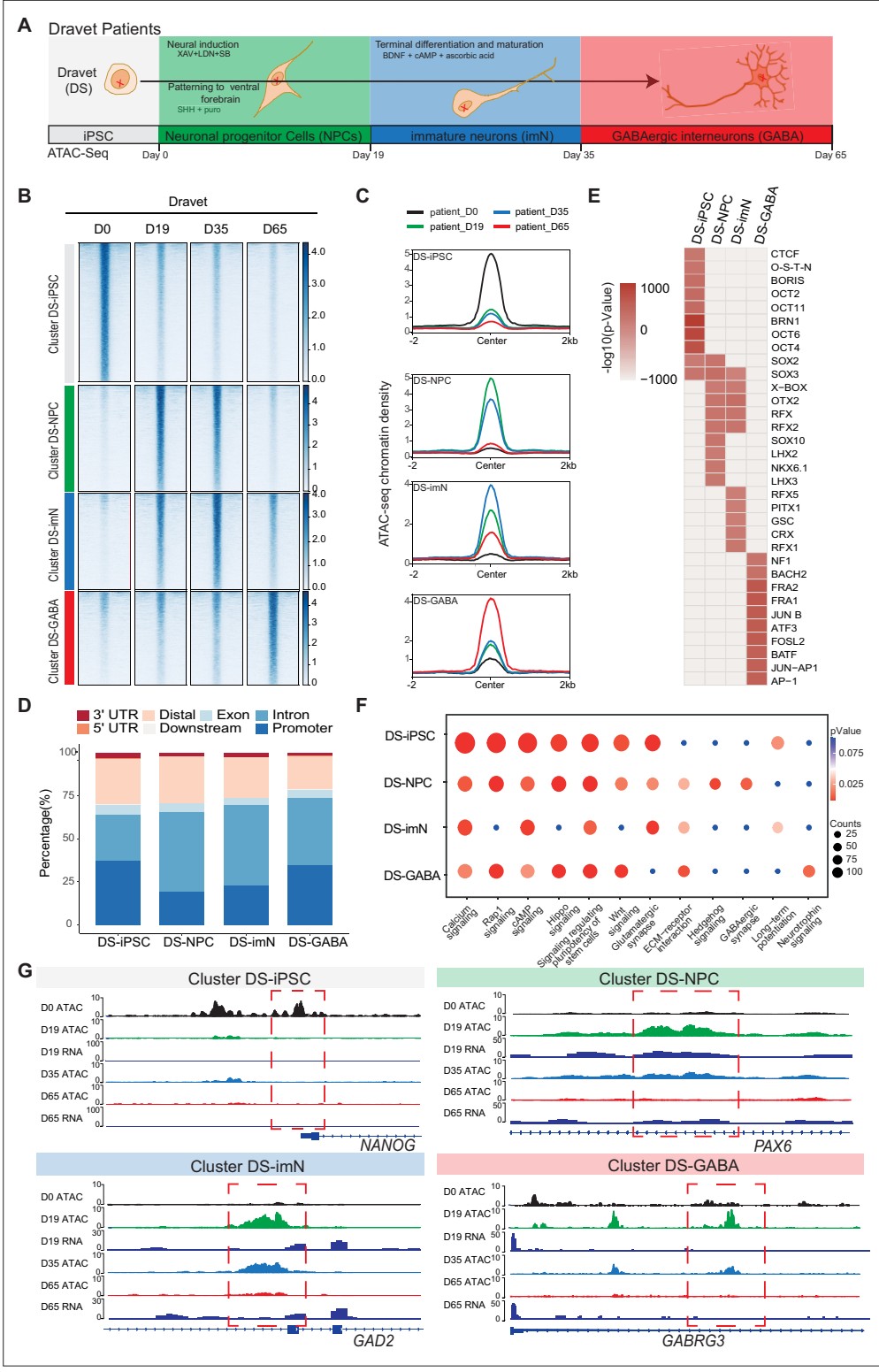

**Figure 2.** Chromatin accessibility dynamics during GABAergic differentiation of Dravet Syndrome patient induced pluripotent stem cells (iPSCs). (**A**) Schematic Illustration of the design for GABAergic differentiation in Dravet syndrome patients. Cells from iPSC differentiation were collected for assay of transposase accessible chromatin sequencing (ATAC-Seq). GABAergic differentiation in Dravet Syndrome patients was grouped into four stages: iPSC (Day 0), NPCs (Day 19), imN (Day 35), GABA (Day 65). (**B**) Heatmap of four clusters identified by time series analysis using temporal changes in chromatin accessibility compared D0 (Day 0) and other points. The signal

*Figure 2 continued on next page*

*Figure 2 continued*

strengths were denoted by color intensities (**C**) Line plots showing chromatin accessibility at cluster-specific regions from each time point. (**D**) Barplot for genomic distribution of differential chromatin accessible regions for each cluster. (**E**) Heatmap for top 10 transcription factors (TFs) enrichment in each cluster. The significance was denoted by color intensities. O-S-E-N=OCT4-SOX2-TCF-NANOG. (**F**) Bubble plot of Kyoto Encyclopedia of Genes and Genomes (KEGG) pathway enrichment for each cluster. p-value and enrichment were indicated. The corresponding comprehensive list of enrichment terms can be found in **Supplementary file 4**. (**G**) Genome browser view showing representative differential chromatin accessible regions at the indicated gene loci. Additionally, RNA-seq data (**Schuster et al., 2019b**) is visualized at D19 and D65.

The online version of this article includes the following figure supplement(s) for figure 2:

**Figure supplement 1.** Quality control of assay of transposase accessible chromatin sequencing (ATAC-Seq) samples during GABAergic interneuron differentiation in Dravet Syndrome patient induced pluripotent stem cells (iPSCs).

**Figure supplement 2.** Chromatin accessibility cluster identification during GABAergic interneuron differentiation in Dravet Syndrome patient induced pluripotent stem cells (iPSCs).

DS-NPC-specific clusters but not in the later appearing clusters specific for DS-imN and DS-GABA (**Figure 2F**). Additionally, the pathway of long-term potentiation, a hallmark for mature GABAergic cells, was specifically enriched in the DS-imN specific cluster but not in the DS-GABA clusters. Furthermore, representative annotated genes for Ctl-iPSC lines, PAX6 and GAD2 in the Ctl-NPC and Ctl-imN clusters, and GABRG3 in the Ctl-GABA cluster, exhibit a different tendency of chromatin accessibility in DS-iPSC clusters (**Figure 2G**), also shown by qPCR analysis (**Figure 2—figure supplement 2E**). The enrichment of TF motifs and predicted pathways relevant for GABAergic interneuron appearing already in DS-NPC, but not in DS-GABA, uncovered a pattern distinct from that in Ctl-iPSC lines, that strongly suggests a perturbed GABAergic development associated with *SCN1A* variants.

## Common chromatin dynamics in Ctl- and DS-iPSC models of GABAergic development

Identification of abnormal chromatin dynamics in GABAergic interneuronal development of DS-iPSC lines requires a thorough comparison with Ctl-iPSC lines. To extract shared dynamic chromatin features in DS-patient and Ctl-iPSC lines, the differentially accessible chromatin peaks specific to each of the four-time points (D0, D19, D35, and D65) within the Ctl and DS patient groups were merged, respectively. The lists of all significant peaks in the Ctl- and DS- iPSC lines, respectively, were subsequently intersected to identify the common peak regions. In total, 12,256 accessible chromatin peaks were identified that show a shared and time-point specific pattern (**Figure 3A**, **Supplementary file 5**).

Next, the genomic annotations, KEGG pathways, and TF motif enrichment for these common peaks were examined (**Figure 3**). A K-mean clustering of read counts from the 12,252 accessible chromatin peaks revealed four common clusters (Com-1 with 6264 peaks, Com-2 with 2071 peaks, Com-3 with 1793 peaks, and Com-4 with 2124 peaks) (**Figure 3A**, **Figure 3—figure supplement 1**). The DS-iPSC and Ctl-iPSC lines show a similar sequencing read count enrichment in the Com-1 cluster and Com-2 clusters at the four time-points whereas slight differences are observed for the Com-3 and Com-4 clusters (**Figure 3A**). Accessible regions in the Com-1 cluster are specifically open at D0 (iPSC) with annotations to TF genes for pluripotency such as *NANOG* and *SOX2*, among others. The Com-1 cluster remains closed at the following three time-points. The Com-2 cluster is closed at D0 but open at D19 (NPC) with accessible peaks at genes such as *MAP2*, *NEUROD1*, and *NEUROG1*, among others (**Figure 3A**).

In both the Ctl and DS-patient groups, accessible regions in the Com-3 cluster are closed at D0 but slightly open at D19 (NPC) and at D35 (imN) and then more closed at D65 (GABA). However, the degree of openness for a large proportion of peaks is much higher in the DS-patient group when compared to the Ctl group. Furthermore, at D65 (GABA) the Com-3 cluster shows accessible regions at NPC-specific genes such as *MAP2* and *ATOH1* but not for regulatory elements of GABAergic neuron-specific genes, such as *ASCL1*, *LHX6*, and *DLX2*, among others. The absence of common open chromatin regions at regulatory elements for GABAergic-specific genes in the Ctl and DS-patient groups supports altered dynamics of accessible chromatin at the late time-point GABAergic development in the model system.

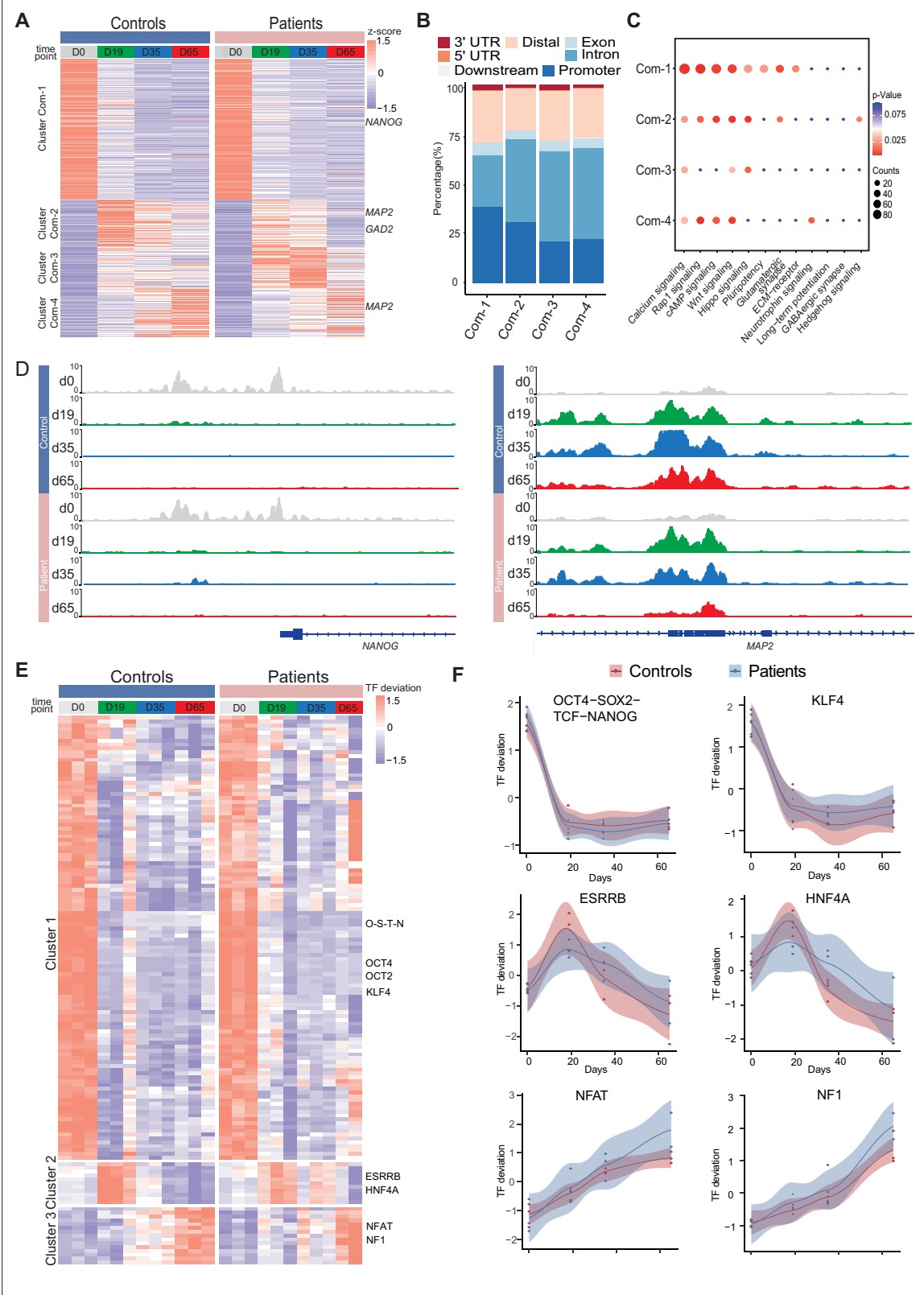

**Figure 3.** Common chromatin accessibility features of control and Dravet Syndrome patient induced pluripotent stem cells (iPSCs) during GABAergic differentiation. (**A**) Heatmap for common chromatin accessible regions detected between control and Dravet Syndrome (DS) group during differentiation. Four clusters were obtained from time series analysis. Representative genes are labeled on the right side of heatmap. (**B**) Barplot for genomic distribution of chromatin accessible regions for each cluster. (**C**) Bubble plot of KEGG pathway enrichment for each cluster. p-value and

*Figure 3 continued on next page*

*Figure 3 continued*

enrichment were indicated. (**D**) Genome browser view showing representative chromatin accessible regions at the indicated gene loci. These genes represent common changing regions between control and patients groups. (**E**) Transcription factors (TFs) deviation Z scores heatmap for unique TF enriched in each cluster. Representative TFs are labeled on the right side of heatmap. (**F**) Representative TFs enrichment (deviation Z score) dynamics at ATAC-Seq peaks in control and Dravet Syndrome (DS) group during differentiation.

The online version of this article includes the following figure supplement(s) for figure 3:

**Figure supplement 1.** Identification of common chromatin accessibility between control and Dravet Syndrome patient induced pluripotent stem cells (iPSCs) during GABAergic interneuron differentiation.

The genomic distribution of accessible peaks with respect to promoters, exons, and introns varies between the Com-clusters (*Figure 3B*). The Com-1 cluster shows a relatively large proportion of accessible chromatin regions from promoter regions (38.2%) when compared to the other clusters (Com-2: 22.0%; Com-3: 20.8%, and Com-4: 30.5%). Conversely, the Com-1 cluster shows a relatively small proportion of accessible regions in intronic regions (25.8%) when compared to Com-2, 3, and 4 clusters. However, the genomic distribution of accessible peaks was very similar in the Com-2 and Com-3 clusters.

Then, KEGG analysis on genes within peaks of the four Com-clusters to understand the common biological features of the Ctl- and DS-iPSC lines was conducted. The enrichment for the pathways of transcriptional regulation of pluripotency was observed, that included *NANOG*, in Com-1 (*Figure 3C*) and of nitric oxide-stimulated guanylate cyclase that included *MAP2* in Com-2 (*Figure 3D*). However, in the Com-3 or Com-4 clusters, characterized by an increasing open chromatin with development, the enrichment in pathways relevant to GABAergic cells was not found (*Figure 3D*). These observations further indicate similarities in the Ctl and DS iPSC-lines in cellular processes up to D19 (NPC), but significant differences at D35 (imN) and D65 (GABA).

Furthermore, the enrichment of motifs for trans-acting transcription factors (TFs) from all common peaks and clustered TFs was examined with a k-means method. The TF enrichment patterns are similar in the Ctl and DS groups, and three TF clusters were observed. Cluster 1, containing pluripotent-specific TFs such as OCT4 and KLF, among others, shows enrichment at iPSC in both the control and DS patient groups (*Figure 3E and F*). Cluster 2, including TF motifs for the NPC-specific TFs ESRRB and HNF4A, is enriched in iPSC (D19) and imN (D35) (*Figure 3E and F*), while some TFs relevant to GABAergic cells, such as NFAT and NF1, are enriched in D65. No GABAergic neuron-specific TFs (e.g. DLX, LHX, ASCL) are observed. Taken together, the data suggest similarities between Ctl- and DS-iPSC lines in accessible chromatin at loci for TFs important for GABAergic induction and early development. However, the Ctl- and DS-iPSC groups do not show similarities in the enrichment of TFs that are critical at later stages of GABAergic maturation.

## Distinct chromatin features in DS-iPSC GABAergic development

A PCA analysis of ATAC-Seq data at D0 demonstrated that profiles of chromatin accessibility are similar in Ctl-iPSC and DS-iPSC-lines at d0 (iPSC) but different at d19 (NPC), d35 (imN), and d65 (GABA) (*Figure 4A*). These data suggest a diverging and disrupted trajectory of GABAergic development in the DS-iPSC model when compared to that in Ctl-iPSC.

To further understand the molecular regulators of dysfunctional GABAergic development in DS-iPSC, the chromatin-accessible peaks were extracted that were mutually exclusive in the Ctl- and DS-iPSC lines at each time point (as described in the Methods section). The peaks were categorized into a Ctl distinct peak list and a DS distinct peak list. In total, the Ctl distinct peak list contained 3311 accessible peaks (0 peaks at D0; 42 peaks at D19; 1490 peaks at D35; and 1779 peaks at D65), while the DS distinct peak list contained 4434 accessible peaks (0 peaks at D0; 13 peaks D19; 1708 peaks D35; and 2713 peaks D65) (*Figure 4B*, *Figure 4—figure supplement 1A*, *Supplementary file 6*). These distinct peaks were located at different genomic regions (*Figure 4C*). Furthermore, KEGG enrichment analysis of the distinct peaks at D65 revealed pathways specific to cortical neurons, such as GABAergic synapse, Glutamatergic synapse, Dopaminergic synapse, and others, in Ctl-GABA but not in DS-GABA (*Figure 4D*, *Figure 4—figure supplement 1B*).

Next, the distinct peak lists of the Ctl- and DS-iPSC lines were combined and the enrichment of TF motifs was calculated using a systematic approach with chromVAR (chromatin variation across regions) (*Schep et al., 2017*). Then, a non-hierarchical clustering of TFs enriched in the distinct groups was

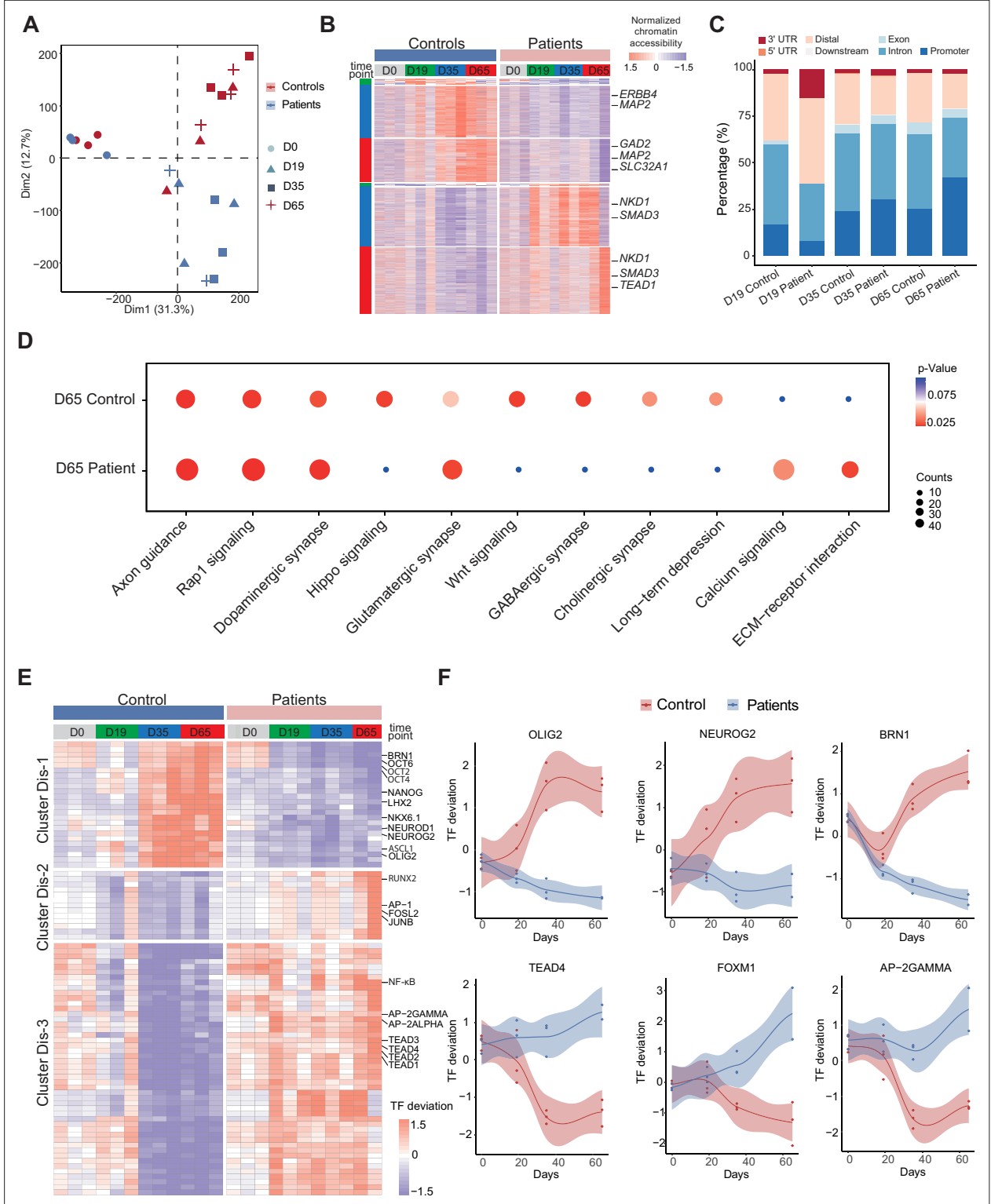

**Figure 4.** Unique chromatin accessibility features of control and Dravet Syndrome patient induced pluripotent stem cells (iPSCs) during GABAergic differentiation. (**A**) Principal component analysis (PCA) plot of all assay of transposase accessible chromatin sequencing (ATAC-Seq) from control and Dravet Syndrome patients during GABAergic differentiation. (**B**) Heatmap showing the differential chromatin accessible peaks for comparison at each time point between the control and Dravet Syndrome (DS) patients. Representative genes are labeled on the right side of heatmap. (**C**) Barplot for genomic features of unique chromatin accessible regions at each time point for the control and DS patients. (**D**) Bubble plot of Kyoto Encyclopedia of Genes and Genomes (KEGG) pathway enrichment at GABAeric interneuron (Day 65) for the control and Dravet Syndrome (DS) patients. p-value and

*Figure 4 continued on next page*

*Figure 4 continued*

enrichment score were indicated. (**E**) Heatmap showing transcription factors (TFs) enrichment at differential chromatin accessible peaks for comparison at each time point between the control and DS patients. Representative TFs are labeled on the right side of heatmap. (**F**) Representative TFs enrichment (deviation Z score) dynamics at ATAC-Seq peaks in control and Dravet Syndrome (DS) group during differentiation.

The online version of this article includes the following figure supplement(s) for figure 4:

**Figure supplement 1.** Identification of unique accessible chromatin region between control and Dravet Syndrome patient induced pluripotent stem cells (iPSCs) during GABAergic interneuron differentiation.

performed and three clusters were identified (Dis-1, Dis-2, Dis-3) (*Figure 4E*). Cluster Dis-1 comprised two subclusters of TFs: Subcluster 1 included the OCT family of TFs (OCT2, OCT4, OCT6) and BRN1, with enrichment restricted to D0 (iPSC) in the DS group whereas in the Ctl group, the enrichment was observed at D0, D35, and D65 but not at D19. Subcluster 2 included the GABAergic-specific TFs OLIGO2, ASCL1, NEUROD1, NEUROG2, NKX6.1, LHX2, and others. These TFs were not enriched at any time-point in the DS group but enriched in the control group at D35 and D65 (*Figure 4F*). In cluster Dis-2, TFs such as JUNB, AP-1, RUNX2, and others were enriched in the DS group from d0 (iPSC) to D65 (GABA). However, in the Ctl group enrichment for the Dis-2 cluster was observed only at D0 (iPSC). Cluster Dis-3 is enriched in both Ctl- and DS-iPSC for TFs of the TEAD family, AP-2, and NF-kappa beta. In the Ctl group, the TFs enrichment increased significantly in NPC and decreased (disappeared) in imN and GABAergic neurons. Interestingly, the enrichment of these TFs in cluster Dis-1 persisted in the DS group with development from NPC to GABA. The absent enrichment of GABA -specific TF motifs at D65 in the DS group brings further support for a disrupted chromatin remodeling during interneuron development.

## VPA induces changes in the chromatin landscape of iPSC derived GABAergic interneurons

VPA is a broad-spectrum anti-epileptic drug commonly used in the treatment of DS (*He et al., 2022*). However, the response to VPA treatment varies considerably among affected individuals (*Brunklaus et al., 2020a*). Prior reports suggest that the drug acts as a histone deacetylase (HDAC) inhibitor and thereby impacts chromatin remodeling (*Ghodke-Puranik et al., 2013*; *Ximenes et al., 2012*). Those studies suggested that chromatin changes induced by VPA in GABAergic cells may uncover processes contributing to therapeutic effects of the drug. Hence, Ctl- and DS-iPSC-derived GABAergic interneurons at D65 with and without VPA supplementation at therapeutic concentrations for six days were cultured, and ATAC-Seq was performed on three sample pairs of Ctl-GABA and on two pairs of DS-GABA (*Figure 5A*, *Figure 5—figure supplement 1A–D*, *Supplementary file 7*). Subsequent PCA analysis of the ATAC-Seq data showed that VPA exposure induced variable changes in accessible chromatin that were seemingly iPSC-line specific (*Figure 5B and C*, *Figure 5—figure supplement 1E*). Then, the analysis was focused on the sample pairs showing the most extensive VPA-induced changes in chromatin accessibility, including two Ctl-GABA pairs (Ctl1B and Ctl7C) and one DS-GABA pair (DD5A; *Figure 5C*). Genomic annotation revealed that chromatin changes induced by VPA were distributed in promoters, introns, and distal regulatory elements and at a variable extent when comparing individual Ctl and DS pairs (*Figure 5—figure supplement 1F*). The regions showing VPA-induced changes in accessible chromatin were distributed in promoters, introns, and distal regulatory elements (*Figure 5D*). The changes in chromatin accessibility after VPA treatment observed in both the Ctl- and DS-GABA may reflect unspecific effects of the drug. To validate the hypothesis on chromatin-specific effects of VPA, the enrichment analysis of differentially accessible peaks for each iPSC-GABA pair was performed. The enrichment in chromatin-relevant pathways in both Ctl and DS pairs was analyzed (*Figure 5E*). The comprehensive analysis of open chromatin in response to VPA in Ctl- and DS-GABA thus suggests drug-induced changes that are unspecific and genome-wide. The observation is in line with the reported unpredictable effects of VPA on seizure frequencies in DS.

Since VPA treatment has an effect on seizure frequencies in a subgroup of affected individuals, whether VPA could 'restore' the chromatin accessibility profile of DS-GABA into the chromatin profile of Ctl-GABA was investigated. The pathways relevant for GABAergic cells became enriched in the chromatin regions that become accessible after VPA treatment in the DS-GABA sample pair (*Figure 5E*). A specific analysis of TF motifs becoming accessible in VPA-treated DS-GABA uncovered enriched

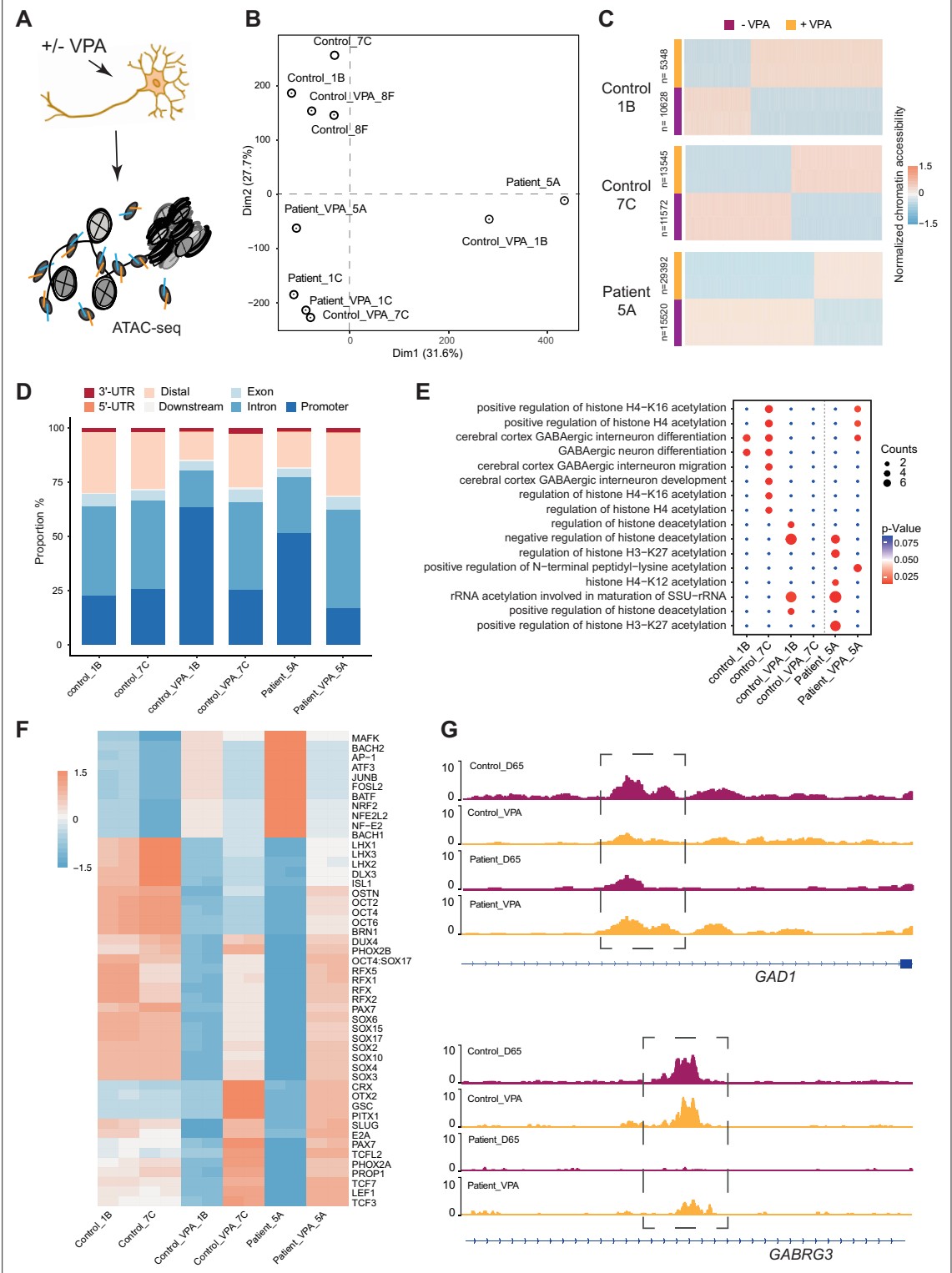

**Figure 5.** Chromatin accessibility response to Valproic acid treatment in GABAergic interneurons. (**A**) Schematic illustration showing measurement of chromatin accessibility response in valproic acid (VPA) treatment of GABAergic interneurons. (**B**) Principal component analysis (PCA) plot of assay of transposase accessible chromatin sequencing (ATAC-Seq) with and without valproic acid (VPA) treatment of GABAergic interneuron in control and Dravet Syndrome patients. (**C**) Heatmap of differential chromatin accessible peaks from the VPA responsible group, consisting of two control samples and one Dravet Syndrome (DS) patient sample. (**D**) Barplot for genomic distribution of VPA responsible accessible regions from differential conditions. (**E**) Bubble plot of Kyoto Encyclopedia of Genes and Genomes (KEGG) pathway enrichment for each condition. p-value and enrichment were

*Figure 5 continued on next page*

*Figure 5 continued*

indicated. (**F**) TF deviation Z score heatmap for top 50 transcription factors (TFs) enriched at responsible chromatin regions from differential conditions. (**G**) Genome browser view showing representative VPA responsible chromatin accessible regions at the indicated gene loci.

The online version of this article includes the following figure supplement(s) for figure 5:

**Figure supplement 1.** Valproic acid treatment reshapes chromatin accessibility of GABAergic interneurons.

motifs for TFs relevant for GABAergic development, such as LHX3, SOX, and the RFX families of TFs (*Figure 5F*). Conversely, Ctl-GABA pairs showed an opposite enrichment pattern for these TF motifs upon VPA treatment. Furthermore, both Ctl- and DS-GABA showed a VPA-induced increase in chromatin accessibility at regulatory regions of GABAergic-specific genes, such as GAD1 and GABRG3 (*Figure 5G*). Taken together, the iPSC model suggests that VPA has the potential to dramatically interfere with and change chromatin structure in GABAergic cells. These changes are likely unspecific and may affect genes important for interneuronal development.

## Discussion

The majority of cases with DS are pharmaco-resistant and there is an urgent need to improve the understanding of the underlying mechanisms for the development of targeted anti-seizure treatments (*Dravet, 2011*; *Brunklaus et al., 2020a*; *Brunklaus et al., 2020b*). Hence, the iPSC-based model of GABAergic development in DS caused by heterozygous *SCN1A* variants was used. The model system recapitulates hallmarks of the disease and it is, therefore, reasoned that it may improve the knowledge on DS-associated epigenetic changes as well as chromatin modifications induced by the commonly used anti-seizure drug VPA (*Schuster et al., 2019b*; *Schuster et al., 2019a*). Specifically, the iPSCs from healthy donors (Ctl) and DS-cases were differentiated towards GABAergic cortical interneurons and characterized the dynamic chromatin conformational changes using ATAC-Seq. The comprehensive analysis reveals chromatin accessibility changes in DS-patient-derived iPSC neurodifferentiation. Both common and distinct chromatin accessibility profiles were identified when comparing iPSC of DS patients and healthy donors at different time points of GABAergic development. As expected, the chromatin profiles of undifferentiated iPSC are similar in the two groups, with accessible peaks for TF genes maintaining a pluripotent state, such as *NANOG* and *POU5F1*. Differentiation towards a GABAergic fate shows that both Ctl and DS iPSC-lines acquire accessible chromatin peaks associated with genes enriched in pathways relevant for interneuronal development (e.g. signaling pathways, GABAergic synapse and long-term potentiation). However, the comprehensive analysis uncovered increased changes in accessible peaks at the early phase (D19) of GABAergic development in DS-patient cells when compared to the control group. At the same time-point (D19), the expression of SCN1A was also detected, providing a link between altered sodium flux and aberrant chromatin dynamics in DS-iPSC derived neurons. The genes associated with the distinct temporal changes of open chromatin in DS-patient cells are enriched in KEGG terms such as GABAergic synapse, Glutamatergic synapse, and Long-term potentiation, suggesting a perturbed and seemingly accelerated initial GABAergic induction before D35. The enrichment in these pathways is lost with further differentiation of DS-derived iPSC-lines and they fail to acquire the chromatin profile of Ctl GABAergic cells at D65. The aberrant chromatin profiles in the iPSC model comply with previous studies on iPSC and mice models of DS showing both transcriptional and electrophysiological changes in interneurons (*Han et al., 2012*; *Schuster et al., 2019b*; *Liu et al., 2013*). In the prior reports, the transcriptional changes emerged predominantly after D19, the ATAC-Seq analysis shows alterations in chromatin structure already at D19. The observation also confirms that epigenetic changes detected by ATAC-Seq precede the transcriptional changes (*Inoue et al., 2019*). Furthermore, the unbiased analysis defines differential accessible chromatin regions and TF motifs associated with heterozygous and pathogenic *SCN1A* variants, bringing further insights into disturbed interneuronal differentiation in DS. In control iPSC lines, the chromatin accessibility of motifs for TFs relevant to drive neurogenesis, such as NEUROG2, OLIG2, BRN1, increase with differentiation whereas the same regions are closing in DS-derived iPSC. On the other hand, motifs of more general TFs, such as TEAD4, FOXM1, AP-2GAMMA, become either accessible or un-accessible with differentiation in control iPSC but remain closed or open in differentiating DS iPSC. The observation is consistent with the previous observations in DS-GABA showing dysregulated expression of *FOXM1* and of genes enriched in pathways for histone modification and cell cycle

pathways (*Schuster et al., 2019b*). Moreover, similar to the *Scn1a*+/- mice model, showing an intact density of cortical and hippocampal GABAergic interneurons despite impaired functions (*Ogiwara et al., 2013*), the immunostainings revealed a comparable number and morphology of DS-patient and Ctl GABAergic neurons. These observations suggest that the proliferative capacity is relatively preserved during GABAergic interneuron development in DS.

Currently, VPA is commonly used as one of the first line drugs in DS (*He et al., 2022*) but the response on seizure frequencies is highly variable among affected cases (*Inoue et al., 2009*; *Dressler et al., 2015*). The mode of action of VPA is not fully understood but it is believed to inhibit histone deacetylases, leading to changes in chromatin states and gene expression (*Jiang et al., 2019*; *Baumann et al., 2021*). Understanding the epigenetic effect of VPA administration on iPSC-derived GABAergic neurons derived from DS patients and healthy donors is critically important to understand the effect of VPA treatment. The GABAergic interneurons exposed to therapeutic concentrations of VPA for 6 days showed extensive changes in accessible chromatin peaks in the cultures. Projection of the chromatin changes using KEGG uncovered pathways that reflect the expected mode of action of VPA, such as *histone acetylation* and *deacetylation* and *rRNA acetylation*, of importance for transcriptional activity.

However, the VPA-induced chromatin modifications in iPSC-GABA varied considerably with biological origin and the accessibility changes linked to GABAergic development were specific or shared for individual iPSC-GABA-lines. For example, the enrichment for LHX1-3 and DLX3 motifs in Ctl-GABA was observed changing from open to closed after VPA exposure. In contrast, DS-GABA show a somewhat opposite enrichment pattern for LHX and DLX binding motifs after VPA exposure that changes from closed to open. In mice, motifs for the LHX and DLX family of TFs are enriched in the late interneuronal development that extends postnatally (*Allaway et al., 2021*). In addition, Ctl-GABA shows a reduction in accessible SOX-binding motifs after VPA treatment whereas DS-GABA shows an opposite response, acquiring a profile more similar to the untreated Ctl-GABA. The SOX-family of TFs has a global impact on both embryonic and adult neurogenesis (*Stevanovic et al., 2021*) and enrichment of SOX binding motifs are enriched in migratory interneurons (*Allaway et al., 2021*). These observations may suggest a mechanism by which VPA, at least in some cases, facilitates accessibility for TFs with beneficial effects on postnatal GABAergic function. Moreover, VPA-treated DS-GABA uncovered enriched changes for TF motifs for AP-1 and BRN1 that made the cells acquire a pattern more similar to that in control lines. It may, therefore, be hypothesized that VPA induces unspecific changes in chromatin structure of iPSC-GABA that, at least in some cases, may lead to an increased accessibility for TFs linked to GABAergic development. Such a variability in response to VPA exposure in the iPSC model is consistent with the unpredictable effects of VPA on seizure-frequencies, even in cases with identical *SCN1A* variants (*Brunklaus et al., 2020a*), mediated by epigenetic factors that are specific to individuals. Finally, the iPSC model of GABAergic development has limitations that should be considered. The model system applies a directed protocol during 65d that may not fully recapitulate the maturation of GABAergic cells in vivo. Functional maturation of GABAergic interneurons occurs postnatally in humans, and models using direct differentiation from iPSC typically do not reach into cellular states comparable to postnatal cell types. More complex neuronal systems, such as 3D organoids from isogenic iPSC lines and in long-term cultures, in combination with single-cell analysis, will be needed to confirm the data. Nevertheless, these findings represent an important step forward to clarify epigenetic changes in inhibitory interneurons that are associated with the progression to DS, adding data to a frame-work of knowledge for the development of targeted and personalized therapies. Further studies are now required to increase the number of independent iPSC lines with pathogenic *SCN1A* variants to validate these findings on epigenetic changes in DS and the effects of anti-seizure drugs such as VPA.

## Materials and methods
### Cell culture, GABAergic differentiation, and sampling

Dravet syndrome iPSC derived GABAergic cells *Schuster et al., 2019b*; *Schuster et al., 2019a* from three patients (DD1C, DD4A, DD5A) and three controls (Ctl1B, Ctl7C, Ctl8F) are all from the previous publication. All cells tested negative for mycoplasma contamination. All procedures were in accordance with the Helsinki convention and written informed consent was obtained from all patients

or their legal guardians. The ethical permit for the research was approved by the regional ethical committee of Uppsala, Sweden (D-numbers 319/2009 and 209/2016).

iPSC lines from three patients (DD1C, DD4A, DD5A) (*Schuster et al., 2019a*) and three controls (Ctl1B, Ctl7C, Ctl8F) (*Sobol et al., 2015*) were cultured on human recombinant laminin LN521 (Biolamina) in Essential-8 medium (Thermo Fisher Scientific) and passaged using Trypl Express (Thermo Fisher Scientific) as described (*Schuster et al., 2019a*). GABAergic interneuron differentiation from iPSCs was performed as previously described (*Schuster et al., 2019b*). The protocol utilizes DUAL SMAD inhibition to induce neurogenesis towards neural stem cells for 10 days, followed by patterning with high levels of sonic hedgehog for 9 days towards cortically fated neuronal progenitor cells (NPC) that mature for 46 days, i.e. a total of 65 days (*Figure 1A*). Neuronal cells at day 65 and onwards are viable as judged by morphological assessment by light microscopy. Differentiation was repeated at least three times per cell line.

Cell cultures were sampled at days 0 (D0), D19, D35, and D65, respectively, by harvesting cells with TryplE and collecting by centrifugation (300 × g, 3 min). Harvested cells were counted and assessed for viability using trypan blue staining and an automated EVE cell counter (NanoEntek). Samples with a viability of >90% were selected for ATAC-Seq library preparation (see below). Additionally, neuronal cell cultures at D65 were treated with valproic acid (VPA; 0,5 mM) for 6 days and subsequently harvested for analysis. Harvested cells were resuspended in 1 ml of 1 x PBS and fixed by adding an equal amount of 2% formaldehyde for 10 min at room temperature. The reaction was quenched by adding 100 ul 2,5 M glycine solution. Fixed cells were collected by centrifugation and washed twice with 1 x PBS. Cells where then resuspended in 1 x PBS, counted using an EVE automatic cell counter (NanoEntek) with Trypan Blue, and subsequently 50,000 cells/sample were processed for library preparation for ATAC sequencing.

## Tn5 transposome assembly

Tn5 transposome assembly was performed by as previously described (*Chen et al., 2016*). Briefly, 2 µM annealed Tn5MErev/Tn5ME-A, 2 µM annealed Tn5MErev/Tn5ME-B, and 2 µM Tn5 transposase (Protein Science Facility at Karolinska Institutet, Stockholm) in dialysis buffer (50 mM HEPES-KOH at pH 7.2, 50 mM NaCl, 0.05 mM EDTA, 0,5 mM DTT, 0.05% Triton X-100, 5% glycerol) with 40% glycerol were mixed, incubated for 1 hr at room temperature and subsequently stored at −20 °C until use. The adaptor oligonucleotides (Tn5MErev: 5'-/Phos/CTGTCTCTTATACACATCT-3', Tn5ME-A: 5'-TCGTCGGCAGCGTCAGATGTGTATAAGAGACAG-3' and Tn5ME-B: 5'-GTCTCGTGGGCTCGGAGATGTGTATAAGAGACAG-3') were synthesized by Integrated DNA Technology. For oligo annealing, equimolar amounts of Tn5MErev and Tn5ME-A or Tn5ME-B and Tn5MErev, respectively, were combined in a PCR tube and annealed using the following program: 95 °C for 5 min, ramp down to 25 °C with −0.1 °C/s increments.

## ATAC-Seq library preparation and sequencing

ATAC-Seq libraries were prepared following *Chen et al., 2016*. In brief, two technical replicates per sample containing 50,000 nuclei each were collected at 500 × g for 5 min at 4 °C. Nuclei were resuspended in lysis buffer (10 mM Tris-Cl, pH 7.4; 10 mM NaCl; 3 mM MgCl2; 0.1% Igepal CA-630) and centrifuged at 500 × g for 10 min. The supernatant was removed and each nuclei pellet was resuspended in tagmentation buffer (25 µL 2 × TD buffer) with 2.5 µl Tn5 transposome and incubated at 37 °C for 30 min. An equal volume of 2 x reverse crosslinking buffer (50 mM Tris-Cl, 1 mM EDTA, 1% SDS, 0.2 M NaCl, 5 ng/ml Proteinase K) was added to each sample and incubated at 65 °C with agitation at 1200 rpm for overnight. DNA was isolated using a MinElute PCR Purification Kit (Qiagen). Sequencing libraries were prepared following described standard protocol (*Buenrostro et al., 2013*). All libraries were sequenced on an Illumina Nova-seq platform at Novogene Europe.

## RNA isolation and quantitative RT-PCR

Total RNA from iPSCs and differentiated cell populations was isolated using a miRNeasy micro kit (Qiagen). 1 µg of total RNA was reverse transcribed into cDNA using a High Capacity cDNA transcription kit (Thermo Fisher Scientific). Expression of marker genes was compared to expression of the two housekeeping genes GAPDH and ACTB. FastStart Universal SYBR Green Master mix (Roche) was used for qPCR using the following primers: ACTB-For: CAGGAGGAGCAATGATCTTGATCT; ACTB-Rev,

TCATGAAGTGTGACGTGGACATC; GAPDH-For, GAAGGTGAAGGTCGGAGTC; GAPDH-Rev, GAAG
ATGGTGATGGGATTTC; SCN1A-For, TGAAGAATCCAGGCAGAAATGC; SCN1A-Rev, TCGAAATG
AACGGAGAACAGA; NANOG-For, CAGCCCCGATTCTCCACCAGTCCC; NANOG-Rev, CGGAAGAT
TCCCAGTCGGGTTCACC; PAX6-For, AACAGACACAGCCCTCACAAACA; PAX6-Rev, CGGGAACT
TGAACTGGAACTGAC; GAD2-For, AGCTGCAGCCTTAGGGATTG; GAD2-Rev, TTGCAAATGTCA
GCGACAGC; GABRG3-For, TCATGGGCCTCAGAAACACC; GABRG3-Rev, CTTGCTGGCGTAGCAT
CTTTT. For all analyses, samples from three independent differentiation cultures were used and each
sample was analyzed in triplicate. Expression was calculated as ΔCT(gene X, day Y)=CT(gene X, day
Y) - CT(AVERAGE (GAPDH/ACTB, day Y)) and fold change was calculated as ΔΔCT(gene X, day Y) =
ΔCT(gene X, day Y) - ΔCT(gene X, day 0) and presented as log2 [2^-(ΔΔCT)] ± SD.

## Immunofluorescence staining

Staining was performed on cells fixed with ice-cold 4% PFA and subsequently permeabilized in blocking
solution (1 x PBS pH 7.4, 1% BSA, 0.1% Triton X-100). Primary antibodies against MAP2 (1:5000;
ab5392, Abcam, Cambridge, United Kingdom), GABA (1:1000; A0310, Sigma, MO, United States),
GAD1 (1:100, MAB5406, Millipore, MA, United States), PV (1:200; ab11427, Abcam, Cambridge,
United Kingdom), SST (1:250, MAB354, Millipore, MA, United States), and DCX (1:100, sc-8066,
Santa Cruz, United States) were used for immunostaining and quantification. Primary antibodies were
allowed to bind overnight separately or in appropriate combinations at 4°C. After washing three times
in 1 x PBS, the secondary antibodies donkey anti-goat IgG AlexaFluor 633 (1:1000, A21082, Thermo
Fisher Scientific, Waltham, MA, United States), donkey anti-rabbit IgG AlexaFluor 568 (1:1000,
A10042, Thermo Fisher Scientific, Waltham, MA, United States) or donkey anti-mouse IgG AlexaFluor
488 (1:1000, A21202, Thermo Fisher Scientific, Waltham, MA, United States) were applied alone or
in appropriate combinations for 1.5 hr at room temperature in the dark. Visualization was performed
on a Zeiss 510 confocal microscope (Carl Zeiss Microscopy, Jena, Germany) using Zen 2009 imaging
software. Image processing was carried out using FIJI software.

## Bioinformatic and statistical analyses

### ATAC-Seq data processing and quality analysis

After the Adapter sequence trimming, the ATAC-Seq sequencing reads were mapped to genome hg38
using bowtie2 (*Langmead et al., 2009*). Mapped paired reads were corrected for the Tn5 cleavage
position by shifting +4/–5 bp depending on the read strand. All mapped reads were extended to
50 bp centered around the Tn5 offset. PCR duplicates were removed using Picard (*Broad Institute,
2014*, MarkDuplicates), and sequencing reads from chromosome M were filtered out. The Peak
calling of each ATAC-Seq library was performed with MACS2 (*Zhang et al., 2008*) with parameters
-f BED, -g hs, -q 0.01, `--nomodel, --shift 0`. Peaks were merged into a matrix using bedtools
merge (*Ramírez et al., 2014*). Raw reads within peaks were normalized using EdgeR's cpm func-
tion (*Robinson et al., 2010*). Log transformation was applied to these normalized peaks to calculate
the Pearson correlation among duplicates. Differential ATAC peaks for clusters were selected using
DESeq2 (*Love et al., 2014*) with the following cutoffs: false discovery rate (FDR)<0.05, |log2 fold
change|>1, and peak average intensity >16. FDR values are Benjamini-Hochberg procedure corrected
per default settings.

### Chromatin accessibility dynamic across differentiation

To avoid capturing the dynamic changes of accessible regions caused by variability across individ-
uals, the dynamic changes of chromatin accessibility cell line by cell line across differentiation were
compared. Subsequently, the common changes observed across different cell lines at each time point
were extracted. Cluster analysis was conducted on both control and patient groups of ATAC-Seq data
at different time points during differentiation (D0, D19, D35, and D65). The raw data were processed
with background correction and normalization using Reads Per Kilobase per Million mapped reads
(RPKM). Clustering analysis was performed separately for the control and patient groups of ATAC-Seq
data using the Mfuzz package (*Futschik and Carlisle, 2005*), with minimum standard deviation and Z
score parameters set at 1 and 0.5, respectively. Clusters were assigned based on the chromatin acces-
sibility patterns of differential chromatin accessible peaks. Subsequently, clusters exhibiting significant
changes were selected for further analysis.

## Annotation of unique ATAC peaks to genes and KEGG pathway analysis

Genomic annotation of each ATAC-Seq peak to its nearest gene for the differential accessible regions was done using ChIPseeker (*Yu et al., 2015*). A gene promoter region was defined as 3 kb upstream and 3 kb downstream of the transcription start site. The peaks were annotated to their nearest gene within a 10 kb distance from the transcription start site (TSS). Subsequently, Clusterprofiler (*Yu et al., 2012*) was used to perform the Kyoto Encyclopedia of Genes and Genomes (KEGG) pathway enrichment analysis, and the results were ranked by false discovery rate (FDR). An FDR of less than 0.05 was set as significant.

## TFs motif enrichment analysis

TF motif enrichment was performed with HOMER *Heinz et al., 2010* using the differential accessible regions as input. For TF binding prediction, chromVAR (*Schep et al., 2017*) was employed. In brief, the Homer vertebrate TF database was used as input for TF motifs in chromVAR, and then TF accessibility deviation values were calculated for each sample across the entire sample set. TF deviations with a threshold greater than 2 were retained, and TF motifs with a positive correlation with one group/cluster were selected to represent that group/cluster. TFs were ranked based on their variability within each group/cluster, and z-scores of deviations from each TF were visualized in a heatmap.

## Statistical analysis

Detailed information for statistical tests performed, p-values, sample sizes, and other descriptive statistics is included in the text and/or in the source data. Parametric tests used: Identification of differential ATAC-Seq peak with false discovery rate (FDR)<0.05, and |log2 fold change|>1. Two-way analysis of variance with Pearson's correlation coefficient.

# Acknowledgements

Research in the lab is funded by the Swedish Research Council (2020–01947 to ND and 2022–00658 to XC), Hjärnfonden (FO2020-0171 and FO2022-0042 to ND), Swedish Cancer Foundation (21 1449Pj, 22 0491 JIA to XC), Stiftelsen Margarethahemmet (to ND) and Sävstaholm Society (to JS), Wallenberg Academy Fellow from Knut and Alice Wallenberg foundation (2023.0046 to XC), Uppsala University and Science for Life Laboratory. The funders played no role in experiment design, data acquisition, and interpretation, or decision to publish.

# Additional information

### Funding

| Funder | Grant reference number | Author |
|---|---|---|
| Vetenskapsrådet | 2022-00658 | Xingqi Chen |
| Vetenskapsrådet | 2020-01947 | Niklas Dahl |
| Hjärnfonden | FO2020-0171 | Niklas Dahl |
| Hjärnfonden | FO2022-0042 | Niklas Dahl |
| Swedish Cancer Foundation | 212119Pj | Xingqi Chen |
| Swedish Cancer Foundation | 220491 | Xingqi Chen |
| Knut och Alice Wallenbergs Stiftelse | 2023.0046 | Xingqi Chen |

The funders had no role in study design, data collection and interpretation, or the decision to submit the work for publication.

## Author contributions
Jens Schuster, Methodology, Writing - original draft, Writing - review and editing; Xi Lu, Yonglong Dang, Joakim Klar, Amelie Wenz, Methodology; Niklas Dahl, Xingqi Chen, Conceptualization, Funding acquisition, Investigation, Writing - original draft, Writing - review and editing

## Author ORCIDs
Jens Schuster ⓘ https://orcid.org/0000-0002-4383-9880
Yonglong Dang ⓘ https://orcid.org/0000-0001-9705-5507
Joakim Klar ⓘ https://orcid.org/0000-0003-4185-7409
Niklas Dahl ⓘ https://orcid.org/0000-0002-8122-0800
Xingqi Chen ⓘ https://orcid.org/0000-0002-5657-2839

Reviewer #2 (Public Review): https://doi.org/10.7554/eLife.92599.3.sa1
Author response https://doi.org/10.7554/eLife.92599.3.sa2

---

# Additional files

## Supplementary files
• Supplementary file 1. Identified differential assay of transposase accessible chromatin sequencing (ATAC-Seq) peaks during GABAergic interneuron differentiation of Ctl-iPSC. Source data for *Figure 1* and *Figure 1—figure supplements 1–3*.

• Supplementary file 2. Kyoto Encyclopedia of Genes and Genomes (KEGG) enrichment for each cluster during GABAergic interneuron differentiation of Ctl-iPSC. Source data for *Figure 1F*.

• Supplementary file 3. Identified differential assay of transposase accessible chromatin sequencing (ATAC-Seq) peaks during during GABAergic interneuron differentiation in Dravet Syndrome patient induced pluripotent stem cells (iPSCs). Source data for *Figure 2* and *Figure 2—figure supplements 1–2*.

• Supplementary file 4. Kyoto Encyclopedia of Genes and Genomes (KEGG) enrichment for each cluster during GABAergic interneuron differentiation in Dravet Syndrome patient induced pluripotent stem cells (iPSCs). Source data for *Figure 2F*.

• Supplementary file 5. Identified common assay of transposase accessible chromatin sequencing (ATAC-Seq) peaks during GABAergic interneuron differentiation between control and Dravet Syndrome patient induced pluripotent stem cells (iPSCs). Source data for *Figure 3* and *Figure 3—figure supplement 1*.

• Supplementary file 6. Identified distinct assay of transposase accessible chromatin sequencing (ATAC-Seq) peaks during GABAergic interneuron differentiation from control and Dravet Syndrome patient induced pluripotent stem cells (iPSCs). Source data for *Figure 4* and *Figure 4—figure supplement 1*.

• Supplementary file 7. Identified differential assay of transposase accessible chromatin sequencing (ATAC-Seq) peaks between with and without valproic acid (VPA) treatment. Source data for *Figure 5* and *Figure 5—figure supplement 1*

• MDAR checklist

## Data availability
The ATAC-Seq data generated for this study have been deposited in the Gene Expression Omnibus (GEO) with the following number: GSE274660. All data generated or analysed during this study are included in the Supplementary files.

The following dataset was generated:

| Author(s) | Year | Dataset title | Dataset URL | Database and Identifier |
|---|---|---|---|---|
| Chen X | 2024 | Epigenetic insights into GABAergic development in Dravet Syndrome iPSC and therapeutic implications | https://www.ncbi.nlm.nih.gov/geo/query/acc.cgi?acc=GSE274660 | NCBI Gene Expression Omnibus, GSE274660 |

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
