## [Editor Report · eLife assessment]

This is a potentially **useful** study that shows changes in the chromatin landscape of GABAergic neurons in induced pluripotent stem cells (iPSCs) derived from both Dravet Syndrome (DS) patients and healthy donors. The strength of the evidence is currently **incomplete** because the authors compared iPSCs from different individuals, rather than isogenic controls. A strategy for minimizing variability across cell lines is used, but the explanation is not complete. The revised manuscript adds RNAseq and qPCR measurements of the expression of the gene SCN1A, however these do not appear to agree, perhaps because of the way the qPCR measurements are normalized, and there is no measurement of Nav1.1, the gene product thought to be responsible for the majority of DS cases. Hence the evidence that there is reduced expression of SCN1A or its gene product is not complete and therefore it is difficult to evaluate whether or not the observed epigenetic changes are causal. The work would potentially be of interest to scientists who study development, developmental disorders, and epigenetic contributions to disease.

---

## [Referee Report · Reviewer #2 (Public Review)]

Summary:

Overall this is an interesting innovative study that examines chromatin accessibility in an inhibitory iPSC model of Dravet Syndrome. The authors detect a potential intriguing development defect in the patient-specific neurons, however the correlation with gene expression or protein abundance is not compelling and the variability of the data is still difficult to determine.

Strengths:

(1) This is a novel and interesting study that aims to investigate the epigenetic changes that occur in a sodium channel model of epilepsy, these are oft ignored, but also an interesting area for future therapeutics.

(2) The paper is well written with good graphics and flow.

(3) With caveats noted below, there is an intriguing developmental defect in GABAergic neuron differentiation in this model. It would be interesting to see how this correlated with the expression of SCN1A, and I was surprised this was not addressed in the manuscript via RNA/protein abundance, nor how the absence of a sodium channel can accelerate differentiation when a priori I might expect the opposite (as less 'neuronal' signal)

(4) There is exploratory analysis that VPA alters chromatin accessibility at an individual-specific level. Though it was not noted if any of the DS patients,

Weaknesses addressed:

(1) Representative images for cell-identity markers are now shown for D19 and D65.

(2) The methods now state that three differentiations were performed.

(3) The authors address a possible role for cell death in data obtained from their cultures by assessing viability with trypan blue staining.

(4) Some features of ATAC signal normalization and enrichment analysis have been better documented.

(5) Some of the variability in key results is better documented.

Weaknesses poorly or not addressed:

(1) Although the authors include prior RNAseq data and report on qPCR measurements for SCN1A (Supp Fig 1)these do not on the surface appear to agree, with the RNAseq showing little apparent difference between patients and controls, while the qPCR seems to show a two-fold difference at D65. This is likely a misleading artifact of normalizing PCR expression to that at D0 when the gene is not expressed but has mildly different low levels in patients and controls. No measurement of the protein product or its function is included. This is a major weakness that casts doubt on the core hypothesis that epigenetic changes play a key causal role in Dravet syndrome.

(2) Although some QC on ATAC is described, QC performed on iPSC lines, i.e. karyotype/CNV analysis and confirmation of genotypes is not described in the paper.

(3) The authors describe a method for trying to diminish variability but do not adequately explain this method or how much variability remains in many of their measures.

(4) Given that VPA would be administered in patients with fully mature inhibitory neurons, it is difficult to determine the biological relevance of these findings.

---

## [Author Response]

The following is the authors’ response to the original reviews.

**Public Review:**
This study used ATAC-Seq to characterize chromatin accessibility during stages of GABAergic neuron development in induced pluripotent stem cells (iPSCs) derived from both Dravet Syndrome (DS) patients and healthy donors. The authors report accelerated GABAergic maturation to a point, followed by further differentiation into a perturbed chromatin profile, in the cells from patients. In a preliminary analysis, valproic acid, an anti-seizure medication commonly used in patients with DS, increased open chromatin in both patient and control iPSCs in a nonspecific manner, and to different degrees in cultures derived from different patients. These findings provide new information about DS-associated changes in chromatin, and provide further evidence for developmental abnormalities in interneurons with DS.Strengths:This is a novel study that aims to investigate the epigenetic changes that occur in a sodium channel model of epilepsy; these changes are often ignored but may be an interesting area for future therapeutics. In general, the flow of the paper is good, and the figures are well-designed. Reply: Thank you for your positive feedback about our work.Weaknesses:The most substantial weakness relates to the observation that DS is often viewed as a monogenic form of epilepsy. It is directly linked to SCN1A gene haploinsufficiency (Yu et al, 2006; Ogiwara et al, 2007). The gene product is Nav1.1, the alpha subunit of voltage-gated sodium channel type I that regulates neuronal excitability. Yet, analysis was conducted at time points of GABAergic interneuron differentiation in which SCN1A is likely not expressed. The paper would be strengthened if SCN1A expression and Nav1.1 protein were examined across the experimental time course. If SCN1A is not yet expressed, this would complicate any explanation of how the observed epigenetic changes might arise. It also seems counterintuitive that the absence of a sodium channel can accelerate differentiation, when, a priori, one might expect the opposite (a 'less neuronal' signal).

Thanks, this is an important point! In our revised manuscript, we have incorporated data on the expression of *SCN1A* at d19 and d65 of GABAergic development in both the control and patient groups. We first retrieved data from our previous RNA-Seq analysis, showing *SCN1A* gene expression in our cells at both d19 and d65. We have now updated our text on the *SCN1A* gene expression in the revised manuscript (Revised Supplementary Figure 1A, revised text Line 108-109). Second, we confirmed the dynamics of *SCN1A* expression by real-time quantitative RT/PCR analysis at four time-pionts of GABAergic development (d0, d19, d35 and d65). Notably, expression of *SCN1A* was detected by qRT-PCR from d19 and the expression increased with differentiation. We have now included this information in the revised manuscript (Revised Supplementary Figure 1B, revised text Line 112).

Related to this, another important limitation of the study is that the controls are cells derived from healthy individuals and not from isogenic lines. The usage of isogenic lines is extremely relevant for every study in which iPSC-derived somatic cells are used to model a disease, but specifically in diseases like DS, in which the genetic background has an ascertained impact on disease phenotype (Cetica et al, 2017 and others). This serious limitation should be considered.

Yes, we fully agree that isogenic and edited patient-derived iPSC would have been the ideal controls. At an early stage we therefore invested considerable time and efforts in order to generate isogenic lines from patientderived iPSC. However, editing of the *SCN1A* variants in patient-derived iPSC turned out unsuccessful after several trials and modifications so we finally turned to iPSC from healthy donors. This is now discussed together with other limitations of our study in the revised manuscript (end of discussion section, lines 499-506).

In addition, the authors should provide data on variability across cell lines and differentiations to help convince the reader that the results can be attributed to genetic defects, rather than variability across individuals.

This is a valuable point. In the revised manuscript, we have now added plots and IF staining from individual samples to give the readers a complete picture on how they are distributed (Revised Supplementary Figure 1C, Revised Supplementary Figure 2, and Revised Supplementary Figure 4).

In the revised manuscript, we incorporated an explanation on the strategy used to compare the two groups (cases *vs*. controls) in more detail. In our analysis, we first compared the dynamic changes of chromatin accessibility cell line by cell line across differentiation. We then extracted the common changes from different cell lines at each time point (Revised text line 152-155, line 226-228). Using this strategy, we extracted the common changes confined to the control and patient groups, respectively. With this approach we avoid to capture the variability across individuals.

Additionally, the authors acknowledge the variability of the differentiations and cell lines, which is commendable, and they attribute this to "possibly reflecting cell line specific and endogenous differences reported previously", but could also have to do with cell death. This is a large confounding factor for ATAC-seq. Certainly, Sup Fig 1C shows lower FrIP scores, consistent with cell death, and there seems to be a lot of death in the representative images. Moreover, the iGABA neurons are very difficult to keep alive, especially to 65 days, without co-culturing with glia and/or glutamatergic neurons. The authors should comment on how much these factors may have influenced their results.

With this point in mind, we re-examined QC of our ATAC-Seq across all samples: As shown in revised

Supplementary Figure 2C and Supplementary Figure 4C, our cutoff for FRiP is 15%, and all of samples have an FrIP of more than 15%. At the later time points (d35 or d65), we did not observe a FRiP <15%. We therefore feel confident that the quality of ATAC-Seq is good enough for downstream analysis and data interpretation.

Regarding the differentiation protocol, we are following a directed protocol of iPSC towards interneurons. The protocol is described in detail by Maroof et al (reference 34) and slightly modified in our lab (described in reference 13). With our modified protocol, GABAergic cells are viable beyond day 65 without the need of co-cultures with astrocyte or microglia. This is also reflected by the electrophysiological activity of interneurons at d65 and at later time points (reference 13). Additionally, our ambition was to obtain a homogeneous cell population for further analysis. Adding other cell types to the cultures would have interfered with downstream processes and a need for cell sorting. Using our protocol, we obtain viable GABA interneurons after up to 100 days in culture. To assess the viability of our cells at the point of sampling (other than by morphological assessment), we used Trypan blue staining and an automated cell counter. Only samples with a viability >90% were processed for ATAC seq. which is a commonly used cut-off for cell viability. We have now modified the method section in the revised version to describe the GABAergic differentiation and sampling (line 519-529).

Finally, changes in gene expression are only inferred, as no RNA levels were measured. If RNA-seq was not possible it would have been good to see at least some of the key genes/findings corroborated with RNA/protein levels vs chromatin accessibility alone, particularly given that these molecular readouts do not always correlate.

In our revised manuscript, we include our recently published RNA-seq performed at d19 and d65. We also correlated the RNAseq and ATACseq data obtained from the same samples. The Pearson correlations between gene expression and chromatin accessibility were within the range 0.49-0.57 (Revised Supplementary Figure 2G, Revised supplementary Figure 4G), which is acceptable according to standard criteria. The results confirmed that the quality of ATAC-Seq is good enough for analysis of expression levels and chromatin openness in key genes. We also added gene expression levels from RNA-seq (d19 and d65) in our revised manuscript (Revised Figure 1G, Revised Figure 2G). Finally, we performed qRT-PCR analysis of key genes in each cluster and the results are now included in the revised version (Revised Supplementary Figure 3E, Revised Supplementary Figure 5E)

Additional Points:(1) Representative images for cell-identity markers for only D65 are shown, and not D0, D19, and D35 though it is stated in the text that this was performed. At a minimum, these representative images should be shown for all lines.

As suggested, we have now added images for cell identity markers of all iPSC lines in the revised version (Revised Supplementary Figure 1C).

(2) What QC was performed on iPSC lines, i.e. karyotype/CNV analysis and confirmation of genotypes?

All iPSC lines used in this study have been fully characterized according to standard and state-of-the art procedures: Expression of pluripotency and stemness genes has been shown by immunostaining, flow cytometry and scorecard analysis; integrity of the genome has been assessed by karyotyping using g-banding; differentiation capacity was characterized using an embryoid body assay in combination with scorecard analysis; and genotypes were verified by Sanger sequencing. Please, see the following publications for full datasets: Schuster et all, Neurobiol Dis 2019, Schuster et al Stem Cell Res 2019, Sobol et al Stem Cells and Development 2015. In our lab, the integrity of iPSC lines are routinely verified using flow cytometry (expression for TRA-1-60 and SSEA4), immunostaining (expression of NANOG, SOX2 and OCT4), Sanger sequencing (targeting variants in SCN1A gene), cell morphology analysis and analysis of mycoplasma by MycoAlert (Lonza).

(3) Were all experiments performed on a single differentiation? Or multiples? Were the differentiations performed with the same type? If not, was batch considered in the analysis?

Thank you for raising this question. The text Material and Methods has been modified as follows, to better describe the differentiation and sampling procedure:

“GABAergic interneuron differentiation from iPSCs was performed as previously described (reference 13). The protocol utilizes DUAL SMAD inhibition to induce neurogenesis towards neural stem cells for 10 days, followed by patterning with high levels of sonic hedgehog for nine days towards cortically fated neuronal progenitor cells (NPC) and subsequent maturation for 46 days, i.e. a total of 65 days (Figure 1A). Neuronal cells at day 65 and onwards are healthy and viable as judged by morphological assessment by light microscopy. Differentiation was performed at least 3 times per cell line.

Cell cultures were sampled at days 0 (D0), D19, D35 and D65, respectively, by harvesting cells with TryplE and centrifugation (300 x g, 3 min). Harvested cells were counted and assessed for viability using trypan blue staining and an automated EVE cell counter (Nano Entek). Samples with a viability of >90% were chosen for ATAC-Seq library preparation (see below).”.

I also assume that technical replicates were merged, and then all three biological replicates were kept for each analysis and outliers were not removed, e.g. Control_D19_8F seems like an example of an outlier.

This is a valuable point. We agree on that there is variability across three health donors and patients, respevtively, but the quality of ATAC-Seq is good after multiple assessment of QC (Revised Supplementary Figure 2B-D). The color code in Supplementary Figure 1C may be mis-leading as the Pearsson correlation of all samples was displayed. Overall, the correlation from all ATAC-seq among replicates are over 0.8. At the same time, we observed that samples at d0 are clustered together, but not at the later time points. We interpret this as related to the cell-line specific plasticity of chromatin dynamic during differentiation. The observation agrees with our results from PCA (Revised Supplementary Figure 2F).

(4) In Figure 1C, it is intriguing that the ATACseq signal gets stronger in imN. One might expect it to be strongest in the iPSCs which are undifferentiated and have the highest levels of open chromatin. Is this a function of sequencing depth, or are all the Y-axes normalized across all time points?

This is another valuable point. Figure 1C present the average chromatin openness for clusters specific regions- not of chromatin openness from the entire genome, which is a reason for why the chromatin openness at

D35 is higher than at other time-points. The genome-wide chromatin openness is presented in revised

Supplementary Figure 2D and we have now updated the figure legend to avoid any potential misunderstanding.

The sequencing depth for each sample is extracted in a similar range. To give the readers a complete picture, we also present the depth of sequencing reads for each sample (Revised Supplementary Figure 2A and Revised Supplementary Figure 4A). The Y-axes of genome browser tracks were normalized, and we added the normalized value in the figures.

(5) In Figure 1F, are these all enriched terms, or were they prioritized somehow?

Yes, the enriched terms are prioritized based on biological meanings, and we have now clarified this in the updated legend of the manuscript. In addition, all enriched terms are now included in revised Supplementary Table 2 and Supplementary Table 4.

(6) In Figure 1G (also the same plots in Fig 2/3), are all these images normalized i.e. there is no scale bar for each track, and do they represent and aggregate BAM/bigwig?

Yes, the genome browser tracks were normalized and we have now revised the figures by adding scale bars.

It would be good to show in supplement the variability across cell lines/diffs - particularly given the variability in the heatmap/PCA - and demonstrate the rigor/reproducibility of these results. This comment applies to all these plots across the 3 figures, particularly as in some instances the samples appear to cluster by individual first and then time point (Sup Fig 3B).

Thanks. We have now revised the figure with plots showing individual samples.

How confident are the authors that these effects are driven by genotype and not a single cell line? In the Fig 3D representation of NANOG, it is very difficult to see any difference between patient and control.

In Figure 3D, we showed *common* chromatin dynamics in the control and patient groups. To avoid any misunderstanding, we have now updated our legend in the revised manuscript.

(7) For the changes in occupancy annotation (UTR/exon/intron etc), are these differences still significant after correcting for variability from cell line to cell line at each time point? I.e. rather than average across all three samples, what is the range? Reply: Revised accordingly.(8) The VPA timepoint is not well-justified. Given that VPA would be administered in patients with fully mature inhibitory neurons, it is difficult to determine the biological relevance. I appreciate that this is a limitation of the model, but this should at least be addressed in the manuscript.

We agree on that our model system of GABAergic interneuron development has limitations and that cells may not fully recapitulate the development and physiology in vivo. Obvious factors to consider in our system are the directed protocol to enrich for GABAergic interneurons and the differentiation time-line restricted to 65d. This is now discussed (lines 499-506).

**Recommendations for the authors:**
(1) The term 'mutation' has been replaced with the term ' pathogenic variant' or likely pathogenic variant depending on the context, please see PMID: 25741868

Thank you for pointing this out. We have replaced all instances of “mutation” with “pathogenic variant” throughout the manuscript.

(2) It is unclear what the nomenclature for sample labelling is in Supplementary Figure 1, e.g. 7C, 8F, 1B.

We apologize for this confusion. There are cell lines names. We labeled all data and images according to cell line name, i.e. control lines: Ctl1B, Ctl7C and Ctl8F; patient lines: DD1C, DD4A, DD5A. To avoid any potential confusion, we have added a note in the revised legend of Supplementary Figure 1B.

(3) Can the authors confirm that the Deseq2 FDR values are Benjamini-Hochberg procedure corrected per default settings? If so, this should ideally be added to methods or legend for clarity

Yes, default settings were used in Deseq2 FDR values, which is added in the method part of revised manuscript.

(4) While it makes sense that the authors present the data in the order of Figure 1, and Figure 2, this actually makes it quite difficult to compare the two datasets, especially for the functional enrichment in the "F" figures. It may be helpful to consider re-organizing the figure order. For instance, for the long-term potentiation signal in the DS-iPSCs, what does this mean in terms of biological relevance? Or maybe Figure 2 needs to be supplementary given that Figure 3 is a more direct comparison.

Thank you for the suggestions. We attempted to reorganize during our revision. We still believe it is easier for the audience to grasp the main message if we organize it according to our current workflow—first presenting an individual differential landscape for controls and patients, and then comparing the common and unique aspects among them.